# Factors and Practices Associated with Self-Medicating Children among Mexican Parents

**DOI:** 10.3390/ph15091078

**Published:** 2022-08-29

**Authors:** Angel Josabad Alonso-Castro, Yeniley Ruiz-Noa, Gissela Cristel Martínez-de la Cruz, Marco Antonio Ramírez-Morales, Martha Alicia Deveze-Álvarez, Raymundo Escutia-Gutiérrez, Candy Carranza-Álvarez, Fabiola Domínguez, Juan José Maldonado-Miranda, Alan Joel Ruiz-Padilla

**Affiliations:** 1Departamento de Farmacia, División de Ciencias Naturales y Exactas, Universidad de Guanajuato, Guanajuato 36050, Mexico; 2Departamento de Ciencias Médicas, División de Ciencias de la Salud, Universidad de Guanajuato, Leon 37320, Mexico; 3Universidad Popular de la Chontalpa, Cardenas 86529, Mexico; 4Departamento de Farmacobiología, Centro Universitario de Ciencias Exactas e Ingenierías, Universidad de Guadalajara, Guadalajara 44430, Mexico; 5Facultad de Estudios Profesionales Zona Huasteca, Universidad Autónoma de San Luis Potosí, Valleys City 79060, Mexico; 6Centro de Investigación Biomédica de Oriente, Instituto Mexicano del Seguro Social, Metepec 74360, Mexico

**Keywords:** self-medication, pediatric, complementary and alternative medicine

## Abstract

Background: Pediatric self-medication is based on the subjective interpretation of symptoms in children by the mother or an adult, the decision to self-medicate is made by a third party. The objective of this work is to provide information on the factors and practices associated with the self-medication of children among parents in Mexico. Methods: A cross-sectional and descriptive study was conducted between June 2020 and December 2021 on Mexican parents with children under 12 years of age. Online questionnaires were completed with sections on sociodemographic characteristics, use of medicines or medicinal plants and their treated symptoms, sources of collection, and their recommendation. Results: A total of 9905 online surveys were completed with representation from the 32 states of Mexico, and the prevalence of self-medication was 49.6% (*n* = 4908). Associated factors were age, having two or more children, children with chronic illnesses, medium educational level, unemployment or employment unrelated to health, medium and high socioeconomic level, and lack of medical security. Respondents self-medicated their children on the recommendation of a family member or friend (55.8%), and own initiative (28%). The most used medication was VapoRub (61.3%), followed by paracetamol (56.9%) and chamomile (33.1%), and the most prevalent symptoms were flu/flow (47.7%) followed by cough (34.2%). The main reasons were perceiving symptoms as not serious (69.9%) and reusing medications (51.9%). Conclusions: There is a high prevalence of self-medication in children in Mexico, mainly associated with children with chronic diseases and families with three or more children.

## 1. Introduction

Self-medication is treating or preventing a disease or symptoms using medications or complementary medicine not prescribed by a physician [1]. Self-medication in children also includes practices such as sharing drugs among members of the family or social group, reusing leftover drugs, not adhering to medical indications, and interrupting or prolonging the dosage or the frequency of administration [2]. The World Health Organization (WHO) has promoted responsible self-medication with over-the-counter (OTC) drugs, emphasizing the importance of information aimed at users to achieve their proper use. Responsible self-medication is related to self-care, including the obtention of information, from a health professional or own experience, for the adequate use of medications [3]. Irresponsible self-medication encompasses mistakes in the diagnosis, incorrect choices in the selection of treatment, dose, route, or schedule of administration, as well as ignoring warnings, risks, or contraindications. This practice carries potential risks and adverse reactions. In addition, irresponsible self-medication might promote drug–drug or herb–drug interactions, leading to hepatotoxicity, nephrotoxicity, the unnecessary expense of public spending, and unnecessary therapeutic duplicity [4].

The metabolism of children is immature. For instance, the content of hepatic microsomal proteins is lower in neonates (26 mg/g) compared to adults (40 mg/g), and the glomerular filtration rate (GFR) in children is 20 mL/min/1.73 m^2^, whereas the GFR in adults ranges from 90 to 120 mL/min/1.73 m^2^ [5]. Therefore, children are more vulnerable and likely to present adverse-drug events, compared to adults [5]. Due to the limited pediatric formulation available on the market [6], off-label use is prevalent, ranging from 40% to 90% [7]. The are many limitations to the use of pediatric medicines. For instance, there are no recommended drugs for children of a certain age, or these medicines are authorized with specific restrictions. In addition, many drugs might be used without following the technical data sheet [8,9,10,11]. Pediatric self-medication, based on the subjective interpretation of the symptoms in children by the mother or an adult, is carried out “by proxy”. The decision of self-medication is made by a third party [12]. This work aimed to provide information about the factors and practices associated with self-medicating children among parents from Mexico. This work gives relevant information to pharmaceuticals, physicians, and the general population about the factors influencing the decision to self-medicating children.

## 2. Results

### 2.1. Demographic Characteristics

A total of 9905 online surveys were completed with representation from the 32 states of Mexico. Most of the respondents were mothers (71.8%) with an average age of 34.64 ± 9.29 years, with one child (41.5%), with children diagnosed with chronic diseases (10.1%), high educational level (50.4%), with employment-related to the health sector (12.7%), with medium socioeconomic level (61.4%), and social security (67.6%) (Table 1). Of all the respondents, 49.6% (*n* = 4908) gave at least one product (allopathic or complementary medicine) to their children without medical indication to treat some discomfort (Table 1).

### 2.2. Associated Factors

The factors associated (*p* < 0.05) with self-medication in pediatrics were age (over 35 years), mothers, having two or more children, children with chronic diseases, high school educational level, unemployed or employment not related to health, medium and high socioeconomic status, and lack of medical security. The factors associated with a high probability of self-medicating children were having children with chronic disease (OR: 1.982 (1.730–2.271)), having three or more children (OR: 1.571 (1.422–1.736)), and being a housewife or unemployed (OR: 1.402 (1.244–1.581)). The use of private insurance was associated (OR: 0.621 (0.621–0.852)) with the probability of not using self-medication in pediatrics (Table 1). Respondents self-medicated their children by recommendation of a relative or friend (55.8%, *n* = 2739), own initiative (28%, *n* = 1376), recommendation of worker in drug store (8.8%, *n* = 431), media (5%, *n* = 243), or consulting an herbalist (2.4%, *n* = 119). A total of 27.1% of the respondents indicated that their children experienced at least one adverse drug reaction. The most frequent adverse reactions were headache (37%), vomiting (24%), dizziness (18%), irritability (6%), and stomachache (5%). Headache (19–25% of the cases associated with adverse reactions) and irritability (3% of the cases) were related to the consumption of loratadine, whereas dizziness (7% of cases), stomachache (2% of cases), and vomiting (14% of cases) were associated with the administration of ibuprofen. Vomiting (5% of cases) and dizziness (5% of cases) were associated with the administration of ambroxol.

### 2.3. Predictors Factors

Multivariate logistic regression analysis was performed to identify significant predictors associated with self-medication in pediatrics, as shown in Table 1. The best predictors of self-medication in children were having children with chronic diseases (AOR: 1.898 (1.652–2.108)), medium socioeconomic level (AOR: 1.404 (1.191–1.656)) or high (AOR: 1.462 (1.212–1.762)), and having three or more children (AOR: 1.387 (1.223–1.572)). Parents with college-postgraduate education were independently associated with the likelihood of self-medication in children (AOR: 1.185 (1.040–1.351)).

### 2.4. Products Used

A total of 24,791 uses of products were reported, with an average of 3.93 ± 3.89 products used by parents for self-medicating their children, 3732 parents (76%) used medications, and 2738 (55.8%) used complementary medicine, of which only 1176 used only complementary medicine. The most used medication was VapoRub (61.3%), followed by paracetamol (56.9%), chamomile (33.1), some type of NSAID (27.7%), and honey (26.9%) (Table 2).

### 2.5. Symptom Treated

The symptoms or discomforts treated with self-medication using allopathic medicine and CAM in pediatrics are described in Table 3. A total of 8535 symptoms are reported with an average of 1.81 + 1.56 for each user. The most prevalent symptoms were flu/nasal flow present in 47.7% of self-medicated children, followed by cough present in 34.2% of users, and indigestion with 17.5% of prevalence (Table 3).

The most commonly used medicinal plants were chamomile, arnica, and Aloe. The symptoms treated with medicinal herbs are described in Table 4. The main way of preparation and administration was infusion with water and oral administration, respectively. The most prevalent symptoms treated with medicinal plants are those related to the respiratory and digestive systems.

### 2.6. Reason for Self-Medication

The main reasons for parents to self-medicate their children were to perceive the children’s symptoms as non-severe (69.9%) and to reuse leftover medications at home (51.9%). Respondents consider that expensive medical consultations and a medical office away from home are not reasons for self-medicating their children (Table 5).

## 3. Discussion

In this study, 49.6% of the respondents self-medicate their children. The prevalence of self-medication in children ranges from 38.5 to 85% in different countries [2,13,14,15]. Except for antibiotics, metronidazole, and some pharmaceutical presentations of ibuprofen, loratadine, and phenylephrine, most of the drugs cited in this report are classified as OTC drugs according to Mexico’s General Health Law [16]. This could explain the high prevalence of self-medication in this study.

Parents mostly used allopathic medicine over CAM for self-medicating their children. This is a common trend in other countries such as Saudi Arabia [17], Finland [18], and Sudan [19]. Among the herbs mentioned by the respondents, none of these are considered toxic for human consumption [20].

VapoRub and acetaminophen were the medications more cited in this study. Acetaminophen was the main medication used for self-medicating children in Saudi Arabia [17], and Sri Lanka [15], whereas in Sudan, Brazil, and Germany the most common drug for self-medicating children were antibiotics, metamizole, and vitamin supplements, respectively [2,13,19]. 

Self-medicating children can result in drug interactions and overdose. Approximately 27.1% of the respondents indicated that their children experienced at least one adverse drug reaction. Hepatotoxicity with acetaminophen in children is reported with doses higher than 75 mg/kg/day [21]. Self-medication with antibiotics may result in bacterial resistance. It is interesting to mention that the prevalence of self-medication with antibiotics was 9.9% in this study. This might be due that antibiotics are obtained only with medical prescriptions in Mexico. Nevertheless, in other studies, the prevalence of self-medication with antibiotics in children in other countries ranges from 6.6 to 36.6% [15,19]. Headache and irritability were the second and fourth most frequent adverse reactions, respectively, due to the consumption of loratadine reported in a study with children from the Netherlands [22]. Gastrointestinal adverse effects were the second most frequent adverse reaction due to the administration of ibuprofen in a study carried out with children from Italy [23]. Similarly, gastrointestinal affections (i.e., stomachache and vomiting) were the main adverse reactions associated with ambroxol in children [24]. The findings found in this study about the adverse reactions reported with loratadine, ibuprofen, and ambroxol corroborate previous studies.

The most common treated symptoms were flu and cough, which are considered non-severe and short-term. This agrees with previous reports [2,13,14], whereas the most common treated symptoms in other studies were fever [17,19] and headache [15].

The most common medicinal herbs were chamomile, arnica, and aloe. Unlike other studies, this work describes what kind of herbs are used for self-medicating children. In Germany, the most prescribed herbal products for children’s health care were ivy leaf, thyme, and eucalyptus, used for treating respiratory diseases [24]. In Nigeria, the most frequently employed plants in pediatrics were *Anogeissus leiocarpus*, *Boswellia dalzielii*, and *Citrus sinensis*, used for respiratory and gastrointestinal diseases [25]. In Uganda, the most cited plants used among children were *Vernonia amygdalina*, *Chenopodium opulifolium*, and *Albizia coriaria*, used for respiratory and gastrointestinal diseases [26]. This study agrees with other studies about the main use of medicinal herbs for treating gastrointestinal and respiratory diseases. Using herbs for health in children is a common practice in different countries. Special attention should be considered to the age of each infant and the dose used in medicinal herbs. Possible adverse reactions and changes in the pharmacokinetics of allopathic medicine associated with drug–herb interactions should be studied. Although none of the herbs mentioned in this study are considered toxic [20], the topical application of eucalypts leads to ataxia, weakness, and unconsciousness in a 6-year-old child, and the topical application of garlic in babies (3 and 6-month-old) resulted in dermatological lesions such as ulcerations, blisters, and non-severe burns [27]. In addition, using eucalyptus oil among children might produce slurred speech, ataxia, and muscle weakness, whereas some Vaccinium and Colocasia species might induce fatal toxicity in children [28].

The main reasons for self-medicating children were non-severe symptoms in children and reusing leftover medicine at home. This agrees with other studies [14,15], whereas the main reasons for self-medicating children in other studies were long waiting times in the clinics and high consultation fees [17,19], and media [2].

A high level of education, being unemployed, and having two or more children were the main factors associated with self-medicating children in Sri Lanka [15], whereas high levels of education and high income were the factors related to self-medicating children in Germany [13]. In Brazil, individuals in the age group 7–18 years with public health care services showed an increased risk for self-medication [2].

This is one of the first studies in Mexico that includes many respondents from all over the country. Some limitations of this study were that most of the respondents were mothers living in urban areas. The results mainly describe the attitudes and factors associated with self-medicating children of mothers. This could be attributed to traditional social roles assigned to women in Latin America and other countries. The results might be different if the study was carried out in rural areas, where health care relies on herbal medicine [29]. It was not obtained how long the medication or complementary product was used. However, most of the medications cited in this study are commonly used for short-period times (3–7 days). Age groups were not considered in this work.

## 4. Materials and Methods

### 4.1. Study Design

This study was cross-sectional and carried out between June 2020 and December 2021 in Mexican parents with children under 12 years old. The appropriate population size for this study was estimated using Raosoft software (Raosoft, Inc. free online software, Seattle, WA, USA). The population of children under 12 years of age in Mexico is 3.5 million in 2020 [30] and it is estimated that approximately 50% practice self-medication; the sample population was 1.75 million residents, the margin error was 1%, the confidence level was 95%, and the response distribution was 50%. A sample size of 9552 was necessary. Only those surveys that were completely compliant were included.

Participants were asked to complete an online questionnaire consisting of sections on sociodemographic characteristics (gender, age, education, employment status, marital status, etc.), the use, without medical recommendation, of different medications or medicinal plants to treat symptoms related to diseases, sources for obtaining herbs or drugs, and who recommended them. Socioeconomic characteristics (number of bedrooms, number of bathrooms, number of occupants, and relationship with the interviewee, etc.) were calculated according to the Mexican Association of Market Research and Public Opinion Agencies [31]. Self-medication was defined as the personal decision of the parents to administer, without consulting a physician, allopathic and/or complementary medicine to their children (12 years old and younger) in the last 6 months. The snowball sampling technique was used to distribute the questionnaire link [32], which includes sending the form link to the authors’ contact lists and encouraging participants to share the link with their contacts. Further, the questionnaire link was then distributed on various social media platforms, including Facebook^®^, Facebook Messenger^®^, and WhatsApp Messenger^®^. The inclusion criteria were parents with children under 12 years of age living in Mexico. All information obtained was kept confidential. The Research Ethics Committee of the University of Guanajuato approved the protocol for this study on 29 May 2020, with the following code CIBIUG-P18-2020.

### 4.2. Data Analysis

The results are reported as means (standard deviations), percentages, and odds ratios (95% CI) when specified. The chi-square test was used to assess associations between self-medication and demographic characteristics. Multivariate logistic regression was performed to assess significant predictors of self-medication among the independent variables using the binary logistic regression test. Results were presented as adjusted odds ratios (AOR) and 95% confidence intervals (CI). *p* values of ≤0.05 were considered statistically significant.

## 5. Conclusions

There is a high prevalence of self-medication practice in children from Mexico, mainly associated with children with chronic diseases and families with three or more children. It is necessary to promote national programs for the rational use of medications to avoid self-medication in Mexican children.

## Figures and Tables

**Table 1 pharmaceuticals-15-01078-t001:** Participant’s characteristics and associations with self-medication.

Characteristic	TOTAL N = 9905	Self-Medication Frequency (*n* (%))		OR ^a^ (95% CI)	AOR ^b^ (95% CI)
		YES *n* = 4908 (49.6)	NO *n* = 4997 (50.4)		
Sociodemographic					
Years (mean ± SD)	34.64 ± 9.29	35.35 + 9.26	33.95 + 9.26	N/A	0.999 (0.991–1.008)
Older than 35 years	4384 (44.3)	2364 (53.9)	2020 (46.1)	1.369 (1.265–1.483) *	1.181 (1.013–1.377) *
18–35 years	5521 (55.7)	2544 (46.1)	2977 (53.9)		
Gender					
Female	7115 (71.8)	3627 (51)	3488 (49)	1.225 (1.122–1.337) *	1.221 (1.116–1.335) *
Male	2790 (28.2)	1281 (45.9)	1509 (54.1)		
Number of children					
1	4109 (41.5)	1802 (43.9)	2307 (56.1)	Ref.	Ref.
2	3266 (33)	1712 (52.4)	1554 (47.6)	1.410 (1.286–1.547) *	1.299 (1.172–1.440) *
3 or more	2530 (25.5)	1394 (55.1)	1136 (44.9)	1.571 (1.422–1.736) *	1.387 (1.223–1.572) *
Chronic disease in children					
Yes	1001 (10.1)	646 (64.5)	355 (35.5)	1.982 (1.730–2.271) *	1.898 (1.652–2.180) *
No	8904 (89.9)	4262 (47.9)	4642 (52.1)		
Education					
Elementary and middle school	1944 (19.6)	940 (48.4)	1004 (51.6)	Ref.	Ref.
High school	2966 (29.9)	1572 (53)	1394 (47)	1.204 (1.074–1.351) *	1.328 (1.174–1.503) *
College-postgraduate	4995 (50.4)	2396 (48)	2599 (52)	0.985 (0.887–1.094)	1.185 (1.040–1.351) *
Place of residence					
Rural	1676 (19.9)	817 (48.7)	859 (51.3)	0.962 (0.866–1.069)	0.978 (0.875–1.092)
Urban	8229 (83.1)	4091 (49.7)	4138 (50.3)		
Employment status					
Housewife/Employed	8177 (82.6)	4146 (50.7)	4031 (49.3)	1.402 (1.244–1.581) *	1.291 (1.137–1.466) *
Unemployed	473 (4.8)	231 (48.8)	242 (51.2)	1.301 (1.053–1.609) *	1.249 (1.003–1.555) *
Related to the health sector	1255 (12.7)	531 (42.3)	724 (57.7)	Ref.	Ref.
Socioeconomic status					
High	3096 (31.3)	1540 (49.7)	420 (50.3)	1.350 (1.147–1.589) *	1.462 (1.212–1.762) *
Middle	6081 (61.4)	3060 (50.3)	3021 (49.7)	1.381 (1.182–1.613) *	1.404 (1.191–1.656) *
Low	728 (7.3)	308 (42.3)	420 (57.7)	Ref.	Ref.
Social security					
Yes	6700 (67.6)	3316 (49.5)	3384 (50.5)	Ref	Ref.
No	2508 (25.3)	1302 (51.9)	1206 (48.1)	1.102 (1.005–1.208) *	1.174 (1.067–1.292) *
Private insurance	697 (7)	290 (41.6)	407 (58.4)	0.621 (0.621–0.852) *	0.730 (0.620–0.858) *

^a^ Univariate logistic analysis based on the results of the chi-square test. ^b^ Multivariate logistic regression analysis based on the results of binary logistic regression test. * Significant variables at *p* value < 0.05. Values are expressed as the mean + SD (quantitative variables) and n (%) qualitative variables. SD = standard deviation; OR, odds ratio; AOR, adjusted odds ratio; 95% CI, confidence interval; N/A, not applicable.

**Table 2 pharmaceuticals-15-01078-t002:** Prevalence in the use of products for self-medicating children.

Products	Used (*n*)	Prevalence Users % *n* = 4908
Medications	3732	76
Vicks VapoRub	3008	61.3
Acetaminophen	2792	56.9
Buscapina compositum	414	8.4
NSAIDs	1358	27.7
Ibuprofen ***	1292	
Ambroxol	784	16
Anti-flu drugs	840	17.1
Loratadine *	675	
Phenylephrine *	126	
Antibiotics	488	9.9
*Amoxicillin* *	431	
Metronidazole	180	3.7
Another	47	1
CAM	2738	55.8
Chamomile	1627	33.1
Honey	1319	26.9
Multivitamin supplements	706	14.4
Arnica	686	14
Aloe	676	13.8
Mint	439	8.9
Guava	434	8.8
Onion	430	8.8
Wormseed	321	6.5
Rice water	287	5.8
Thyme	255	5.2
Others	931	19
Palo azul (wood blue) *	144	
Peppermint *	78	
Lemon *	77	
Garlic *	71	
Ginger *	67	
Mexican mullein *	61	
Eucalypt *	53	

* Refers to the most frequently used product in its subgroup. CAM: complementary and alternative medicine.

**Table 3 pharmaceuticals-15-01078-t003:** Symptoms treated with self-medication in pediatrics with allopathic and CAM.

Symptom	Treated (*n*)	Prevalence Users % *n* = 4908
Flu	2339	47.7
Cough	1679	34.2
Indigestion	857	17.5
Diarrhea	853	17.4
Fever	705	14.4
Wounds	567	11.6
Insomnia	302	6.2
Insect bites	252	5.1
Vomiting	213	4.3
Dermatitis	227	4.6
Urinary infection	110	2.2
Another	431	8.8

**Table 4 pharmaceuticals-15-01078-t004:** List of medicinal plants mentioned by the informants.

Common Name of Medicinal Plant	Way of Preparation/Plant Part Used/Application	Symptom
Chamomile	Infusion of whole plant in water, oral administration	Indigestion, insomnia, and flu
Arnica	Maceration of whole plant in ethanol, topical administration	Wounds
Aloe	Decoction of whole plant, topical administration	Wounds, insect bites, and dermatitis
Mint	Infusion of aerial parts, oral administration	Flu, fever, cough, flu, and vomiting
Guava	Infusion of leaves in water, oral administration	Diarrhea, indigestion
Onion	Infusion of bulb in water, oral administration	Cough
Wormseed	Infusion of leaves and stem in water, oral administration	Indigestion
Rice water	Infusion of seeds in water, oral administration	Diarrhea and indigestion
Thyme	Infusion of whole plan in water, oral administration	Flu and indigestion
Palo azul (wood blue)	Infusion of bark in water, oral administration	Urinary infection
Peppermint	Infusion of leaves and branches, oral administration	Vomiting, cough, and flu
Lemon	Infusion of leaves in water, oral administration	Cough and flu
Garlic	Infusion of bulb in water, oral administration	Cough
Ginger	Infusion of roots, oral administration	Cough
Mexican mullein	Infusion of flowers in water, oral administration	Flu, fever, cough, and flu
Eucalypt	Infusion of leaves in water, oral administration	Cough and flu

**Table 5 pharmaceuticals-15-01078-t005:** Reason for practicing self-medication.

Reason		*n* = 4908 (%)	
	Disagree	Neutral	Agree
Expensive medical consultation	1635 (33.3)	2262 (46.1)	1011 (20.6)
The medical office is away from home	1938 (39.5)	2082 (42.4)	888 (18.1)
Expensive drugs	1293 (26.3)	2078 (42.3)	1537 (31.3)
Reusing leftover drugs	936 (19.1)	1426 (29.1)	2546 (51.9)
Preference for natural medicine	803 (16.4)	2389 (48.7)	1716 (35)
Non-severe symptoms	321 (6.5)	115 (23.5)	3432 (69.9)

## Data Availability

Not applicable.

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
