# Peer review of "Factors and Practices Associated with Self-Medicating Children among Mexican Parents"

_pharmaceuticals, 2022, doi:10.3390/ph15091078_

Round 1

Reviewer 1 Report

Despite of interesting subject, this manuscript lacks methodological details essential to reproduce the study, such as:

1) Is it unclear what the operational concept for self-medication used in this study?

2) Recall period used to measure self-medication. Different recall periods (48 h, one week, one month, etc.) can generate different prevalences.

3) Information on the sampling (The investigators sent the electronic questionnaire to whom? Did they have any initial cadastre? What criteria were used to send the questionnaire to potential web survey participants?)

 4) Data on initial sample size, response rate, refusals, etc.

5) In the analysis, possible confounding factors were not controlled, leading to spurious results, and compromising the discussion. Multivariable regression is recommended.

6) In the method, they mention that they investigated who indicated the drugs/plants, but no results are shown about this.

Author Response

1) Is it unclear what the operational concept for self-medication used in this study?

The following information was included in the manuscript:

Materials and Methods section:

Self-medication was defined as the personal decision of the parents to administer, without consulting a physician, allopathic and/or complementary medicine to their children (12 years old and younger) in the last 6 months.

2) Recall period used to measure self-medication. Different recall periods (48 h, one week, one month, etc.) can generate different prevalences.

The following information was included in the manuscript:

Materials and Methods section:

Self-medication was defined as the personal decision of the parents to administer, without consulting a physician, allopathic and/or complementary medicine to their children (12 years old and younger) in the last 6 months.

3) Information on the sampling (The investigators sent the electronic questionnaire to whom? Did they have any initial cadastre? What criteria were used to send the questionnaire to potential web survey participants?)

The following information was included in the manuscript:

Materials and Methods section:

The snowball sampling technique was used to distribute the questionnaire link [33], which includes sending the form link to the authors' contact lists and encouraging participants to share the link with their contacts. Also, the questionnaire link was then distributed on various social media platforms, including Facebook®, Facebook Messenger®, and WhatsApp Messenger®.

[33] Reuben RC, Danladi MMA, Saleh DA. Knowledge, attitudes and practices towards COVID‐19: an epidemiological survey in North‐Central Nigeria. J Community Health [Internet]. Springer; 2020;1‐14. doi: 10.1007/s10900-020-00881-1 

 4) Data on initial sample size, response rate, refusals, etc.

The following information was included in the manuscript:

Materials and Methods section:

The appropriate population size for this study was estimated using Raosoft software (Raosoft, Inc. free online software, Seattle, WA, USA). The population of children under 12 years of age in Mexico is 3.5 million in 2020 [31] and estimated that approximately 50% practice self-medication, the sample population was 1.75 million residents, the margin error was 1%, the confidence level was 95%, and the response distribution was 50%. A sample size of 9552 was necessary. Only those surveys that were completely compliant were included.

5) In the analysis, possible confounding factors were not controlled, leading to spurious results, and compromising the discussion. Multivariable regression is recommended.

We have added a column of logistic regression results to Table 1 and the following information was included in the manuscript:

Section 4.2

Multivariate logistic regression was performed to assess significant predictors of self-medication among the independent variables using the binary logistic regression test. Results were presented as adjusted odds ratios (AOR) and 95% confidence intervals (CI). P values of ≤ 0.05 were considered statistically significant.

Results section:

The following section was included in the manuscript:

2.3. Predictors factors

Multivariate logistic regression analysis was performed to identify significant predictors associated with self-medication in pediatrics, as shown in Table 1. The best predictors of self-medication in children were having children with chronic diseases [AOR: 1.898 (1.652 – 2.108)], medium socioeconomic level [AOR: 1.404 (1.191 – 1.656)] or high [AOR: 1.462 (1.212 – 1.762)] and having three or more children [AOR: 1.387 (1.223 – 1.572)]. Parent`s College-postgraduate education was independently associated with the likelihood of self-medication in children [AOR: 1.185 (1.040 – 1.351)].  

6) In the method, they mention that they investigated who indicated the drugs/plants, but no results are shown about this.

Section 2.2 describes this information:

Respondents self-medicated their children by recommendation of a relative or friend (55.8%, n=2739), own initiative (28%, n= 1376), recommendation of worker in drug store (8.8%, n=431), media (5%, n= 243), or consulting an herbalist (2.4%, n= 119).

Reviewer 2 Report

The article is interesting, but it does contain some inaccuracies. They concern:

title: should be a bit shorter, in its current form it is quite illegible.

This is also related to the suggestion for keywords, which are too many and do not fully reflect the problem of the topic. In addition, these should be phrases, the most important keys that the reader can use to find the article in search engines. The authors are requested to correct these inaccuracies.

Abstract:

There is no information in this part of the article about the control group and on which group the research was conducted. The authors should present the characteristics of the group / groups of respondents that have been analyzed for self-treatment. There is no specific summary of the overall research that is the subject of this article. Was it a survey? Please complete this.

Introduction: It is quite short and laconic. It does not contain information that would introduce the reader to the meaning of the experiment performed with regard to relevant research regarding the scope of the topic.

The authors did not indicate the novelty of their experiment. What is the difference between the present research and the previously published data from other regions of the world. So please outline the background around the analyzed problem, especially.

Chapter: 2. Results and Discussion and 3. Discussion

Here you should separate these chapters so that they are separate, for example:

2. Results

3. Discussion

Then chapter 5. Conclusions

The authors are asked to formulate specific 2-3 conclusions that emerge from this study.

The methodology is not clear, please give the group of respondents. It is not known on which group the research was conducted.

Author Response

The article is interesting, but it does contain some inaccuracies. They concern:

title: should be a bit shorter, in its current form it is quite illegible.

Dear reviewer, the title contains 10 words (85 characters with spaces). We consider that this title is legible and concise for readers. Thank you for your suggestion.

This is also related to the suggestion for keywords, which are too many and do not fully reflect the problem of the topic. In addition, these should be phrases, the most important keys that the reader can use to find the article in search engines. The authors are requested to correct these inaccuracies.

In this journal, the instructions for authors indicate that: “keywords are specific to the article, yet reasonably common within the subject discipline”. The keywords used in this study meet the criteria established by the journal.

Abstract:

There is no information in this part of the article about the control group and on which group the research was conducted. The authors should present the characteristics of the group / groups of respondents that have been analyzed for self-treatment. There is no specific summary of the overall research that is the subject of this article. Was it a survey? Please complete this.

This is a cross-sectional and descriptive study. There is no control group. The abstract indicates the following: “Online questionnaires were completed with sections on sociodemographic characteristics, use of medicines or medicinal plants and their treated symptoms, sources of collection, and their recommendation”.

The characteristics of the respondents are mentioned in section 2.1. This information is too long to show in the abstract section.

Introduction: It is quite short and laconic. It does not contain information that would introduce the reader to the meaning of the experiment performed with regard to relevant research regarding the scope of the topic.

In this journal, the instructions for authors indicate that: “The introduction should briefly place the study in a broad context and highlight why it is important”. The information presented in the introduction section meets this criterion.

The authors did not indicate the novelty of their experiment. What is the difference between the present research and the previously published data from other regions of the world. So please outline the background around the analyzed problem, especially.

The following information was added in the manuscript to indicate the novelty of the study:

This is one of the first studies in Mexico that includes many respondents from all over the country.

The discussion section compares information of similar studies carried out in other countries and the present study.

Chapter: 2. Results and Discussion and 3. Discussion

Here you should separate these chapters so that they are separate, for example:

  1. Results
  2. Discussion

The sections were separated

Then chapter 5. Conclusions

The authors are asked to formulate specific 2-3 conclusions that emerge from this study.

Two specific conclusions are included in this manuscript:

1) There is a high prevalence of self-medication practice in children from Mexico, mainly associated with children with chronic diseases and families with three or more children.

2) It is necessary to promote national programs for rational use of medications to avoid self-medication in Mexican children.

The methodology is not clear, please give the group of respondents. It is not known on which group the research was conducted.

The methodology section was improved by adding the following information suggested by the reviewers 1 and 3:

The appropriate population size for this study was estimated using Raosoft software (Raosoft, Inc. free online software, Seattle, WA, USA). The population of children under 12 years of age in Mexico is 3.5 million in 2020 [31] and estimated that approximately 50% practice self-medication, the sample population was 1.75 million residents, the margin error was 1%, the confidence level was 95%, and the response distribution was 50%. A sample size of 9552 was necessary. Only those surveys that were completely compliant were included.

Self-medication was defined as the personal decision of the parents to administer, without consulting a physician, allopathic and/or complementary medicine to their children (12 years old and younger) in the last 6 months. The snowball sampling tech-nique was used to distribute the questionnaire link [33], which includes sending the form link to the authors' contact lists and encouraging participants to share the link with their contacts. Also, the questionnaire link was then distributed on various social media platforms, including Facebook®, Facebook Messenger®, and WhatsApp Messenger®.

Multivariate logistic regression was performed to assess significant predictors of self-medication among the independent variables using the binary logistic regression test. Results were presented as adjusted odds ratios (AOR) and 95% confidence inter-vals (CI). P values of ≤ 0.05 were considered statistically significant.

Reviewer 3 Report

The introduction lines 55-67 should be updated with more recent data and exemplified with pharmacokinetic characteristics specific to children. Examples of possible adverse drug reactions encountered in your study as well as drug interactions and herbal products mentioned in the study should be mentioned. They are necessary for the argumentation of the study. Thus, the percentages recorded for the use of medicines or herbal products may show the danger of using them without medical supervision.

Appreciation

The large number of data and associated with socioeconomic aspects.

Disadvantages

The study provides a statistic of  data from the urban environment (aspect mentioned by the authors). If it could be registered under the control of a medical staff (doctor, pharmacist) who could provide important details related to efficacy and safety, the data would have a high degree of objectivity.

Author Response

The introduction lines 55-67 should be updated with more recent data and exemplified with pharmacokinetic characteristics specific to children.

The following sentence was included in the introduction section:

For instance, the content of hepatic microsomal proteins is lower in neonates (26 mg/g) compared to adults (40 mg/g), and the glomerular filtration rate (GFR) in children is 20 ml/min/1.73 m2, whereas the GFR in adults ranges from 90 to 120 ml/min/1.73 m2 [5].

The following references were included in the manuscript for updating the information:

Moulis F, Durrieu G, Lapeyre-Mestre M. Off-label and unlicensed drug use in children population. Therapie (2018) 73(2): 135–149. https://doi.org/10.1016/j.therap.2018.02.002

Wilfond BS. Pediatric Drug Labeling and Imperfect Information. Hastings Cent Rep. (2020) 50(1): 3. https://doi.org/10.1002/hast.1074

Gore R, Chugh PK, Tripathi CD, Lhamo Y, Gautam S. Pediatric Off-Label and Unlicensed Drug Use and Its Implica-tions. Curr Clin Pharmacol. (2017) 12(1): 18–25. https://doi.org/10.2174/1574884712666170317161935

Examples of possible adverse drug reactions encountered in your study as well as drug interactions and herbal products mentioned in the study should be mentioned. They are necessary for the argumentation of the study. Thus, the percentages recorded for the use of medicines or herbal products may show the danger of using them without medical supervision.

The following information was included in the manuscript:

Section 2.2.

The most frequent adverse reactions were headache (37%), vomiting (24%), dizziness (18%), irritability (6%), and stomachache (5%). Headache (19-25% of the cases associated with adverse reactions) and irritability (3% of the cases) were related to the consumption of loratadine, whereas dizziness (7% of cases), stomachache (2% of cases) and vomiting (14% of cases) were associated with the administration of ibuprofen. Vomiting (5% of cases) and dizziness (5% of cases) were associated with the administration of ambroxol

Discussion section

Headache and irritability were the second and fourth most frequent adverse reactions, respectively, due to the consumption of loratadine reported in a study with children from the Netherlands [23]. Gastrointestinal adverse effects were the second most frequent adverse reaction due to the administration with ibuprofen in a study carried out with children from Italy [24]. Similarly, gastrointestinal affections (i.e., stomachache and vomiting) were the main adverse reactions associated with ambroxol in children [25]. The findings found in this study about the adverse reactions reported with loratadine, ibuprofen, and ambroxol corroborate previous studies.

Appreciation

The large number of data and associated with socioeconomic aspects.

Thank you for your comments.

Disadvantages

The study provides a statistic of data from the urban environment (aspect mentioned by the authors). If it could be registered under the control of a medical staff (doctor, pharmacist) who could provide important details related to efficacy and safety, the data would have a high degree of objectivity

Senior pharmacists participated in this study. We used an online questionnaire due to the pandemic situation. After the acceptance for publication of this manuscript, we will publish divulgation articles, accessible to the general population, about self-medication in children. The findings will be spread by the media.
